# Concern for Highly Pathogenic Avian Influenza Spillover into Cetaceans

**DOI:** 10.3390/v17121536

**Published:** 2025-11-24

**Authors:** Teresa Pérez-Sánchez, José Carlos Báez, Carolina Johnstone

**Affiliations:** 1Centro Oceanográfico de Málaga, Instituto Español de Oceanografía, Consejo Superior de Investigaciones Científicas (IEO-CSIC), 29002 Málaga, Spain; teresa.perez@ieo.csic.es (T.P.-S.); josecarlos.baez@ieo.csic.es (J.C.B.); 2Departamento de Microbiología, Facultad de Ciencias, Universidad de Málaga (UMA), 29010 Málaga, Spain; 3Instituto Iberoamericano de Desarrollo Sostenible (IIDS), Universidad Autónoma de Chile, Temuco 4810101, Chile

**Keywords:** dolphin, host adaptation, marine epizootic, marine mammal, porpoise, zoonotic risk, viral transmission, whale

## Abstract

Influenza A virus (IAV) has a wide range of avian and mammalian hosts, leading to disease outbreaks and increasing the risk of panzootics and pandemics. Subtype H5N1 of clade 2.3.4.4b is causing the current high pathogenicity avian influenza (HPAI) panzootic. Environmental changes are fuelling the spread of HPAI H5N1 in wildlife worldwide, with occasional spillover events from seabirds to cetaceans. Sampling difficulties and limited tests available for diagnosis are a challenge to cetacean virology research. Understanding the risk of HPAI outbreaks in cetaceans requires a comprehensive examination of events of IAV infection. Documented cases relate to IAV subtypes H1N3, H13N2, H13N9, and H5N1 and have been reported in cetaceans sampled in the Pacific, Atlantic, and Arctic Oceans. The number of H5N1 IAV isolated from cetaceans is increasing and affects six host species of the families *Delphinidae* and *Phocoenidae* of the suborder *Odontoceti*. The analysis of 40 molecular markers of viral adaptation to mammals in 21 H5N1 cetacean isolates reveals mutations are present in three viral proteins: hemagglutinin (HA), polymerase basic protein 2 (PB2), and nucleoprotein (NP). Phylogenetic analysis of HA and PB2 sequences isolated from cetaceans and co-occurring cases in seabirds and marine mammals do not support sustained transmission of the virus between cetaceans. IAV H5N1 appears to be reaching cetaceans after spillover from seabirds and other marine mammals. Increasing worldwide surveillance of IAV infection of cetaceans is crucial, as these marine mammals are sentinel species for human pandemic preparedness and key species for marine biodiversity conservation and ecosystem health.

## 1. Introduction

Influenza A virus (IAV) causes avian influenza (AI) epizootics and endangers wildlife, human health, and agricultural production [1,2]. IAV is an enveloped virus that harbours a negative-sense single-stranded segmented RNA [3]. Its genome is organised in eight viral ribonucleoproteins (vRNP) that encode 17 proteins, some of which are essential for viral replication [4]. Polymerase basic protein 2 (PB2), polymerase basic protein 1 (PB1), and polymerase acidic protein (PA) work together and compose the IAV polymerase [4]. The matrix protein 1 (M1) shapes the matrix layer under the envelope of the viral particle to give structural support, nucleoprotein (NP) is part of vRNP, whereas hemagglutinin (HA) and neuraminidase (NA) are responsible for the budding of new virions [5].

Surface glycoproteins HA and NA are used to classify IAV subtypes. There are 18 different HA and 11 NA, of which H1 to H16 and N1 to N9 have been identified from aquatic birds [6]. IAV evolve through either accumulation of mutations (antigenic drift) or through reassortment of viral genomic segments often originating from different hosts (antigenic shift) [7]. HA binds to sialic acid, a nine-carbon monosaccharide that is the receptor of IAV in the host cell, through N-acetylneuraminic acid (Neu5Ac), preferably α2,3- or α2,6-linked Neu5Ac [8,9]. In birds, especially waterfowl, IAV causes AI that can be low or highly pathogenic (LPAI and HPAI, respectively), based on mortality rates in chickens [10]. LPAI virus (LPAIV) have a monobasic protease cleavage site in HA, in contrast to HPAI virus (HPAIV), that mainly have HA subtypes H5 or H7 harbouring a polybasic cleavage site that generates virulent strains, processed by ubiquitous proteases that lead to a fast-spreading deadly disease [11,12], at least in birds and poultry [13]. On the other hand, LPAIV causes from no signs to mild symptoms due to its replication happening primarily in the avian gastrointestinal tract [13].

Subtype H5N1 of clade 2.3.4.4b is causing the current AI panzootic, raising concern for animal and global health [2]. The H5N1 that appeared for the first time in China in 1996 (goose/Guangdong (gs/Gd) lineage) and became the first strain to create a maintained transmission in domestic poultry [6]. Its activity has significantly increased worldwide since 2021, leading to an increase in mass mortality events in poultry and wild birds and outbreaks in mammals, marine ones included [1]. Even though its expansion happened in 2021, its presence has been detected in a huge range of mammal species since 2003, generating concern as its adaptation to mammals implies an important threat for wild mammal species and human health [14]. The frequent transmission of this HPAIV to mammals is alarming as it raises the risk of the virus to potentially evolve and spread efficiently among them [15], with devastating consequences in biodiversity and conservation when arriving to naïve populations. This mammal-to-mammal transmission has been reported for domestic animals such as minks in Spain [16] or dairy cows in the USA [17], leading to sporadic cases in humans [18]. The ongoing H5N1 panzootic has also caused important outbreaks in wild mammals with increased mortality events in marine species like seals or sea lions [15,19,20]. Viral molecular features and fitness determinants have been deeply characterised [7,21,22], and recently, the European Food Safety Authority (EFSA) has proposed a list of mutations to follow for preparedness, prevention, and control of zoonotic AI [22].

IAV H5N1 is positioned as the second main cause of mass mortality events in marine mammals, although it does not seem to be of concern when it comes to cetacean mammals such as dolphins, porpoises, or whales, at least for now [23]. To understand the risk of HPAI outbreaks in cetaceans we performed a comprehensive review of documented cases of IAV infection of cetacean species (Table 1). Different circulating IAV strains have spread to cetaceans around the globe, causing mainly respiratory and neuronal problems but never massive mortality events. The first registered case reported LPAI H1N3 in 13 positive lung samples and one liver sample collected from 72 common minke whales (*Balaenoptera acutorostrata*) captured with a whaling flotilla [24]. LPAI subtypes H13N2 and H13N9 were detected and isolated in 1984 from a long-finned pilot whale on the coast of New England [25]. Interestingly, H13N2 was detected in both samples extracted (lungs and hilar nodes), while H13N9 appeared only in the hilar node samples. Several cases of IAV infection reported in cetaceans from 1990 to 2006 were unable to determine the subtype [26,27]. For instance, the first cases of IAV reported in belugas were determined by detection of antibodies specific to IAV in serum or blood through ELISA [28] and also by RT-PCR [27]. The currently circulating HPAI subtype H5N1 clade 2.3.4.4b is the only subtype of IAV isolated from cetaceans since 2022 [24,25,26,27,28,29,30,31,32,33,34,35,36,37,38,39]. The method of viral diagnosis of IAV cases reported in cetaceans is usually not informed, and according to symptomatology, the majority of IAV cases were reported in cetaceans that were dying or already dead. General symptoms included neurological disorders that caused the animal to swim in circles or unable to dive [20,25,34]. Neuronal necrosis, inflammation of brain and meninges, or haemorrhagic lungs were some of the characteristics reported when tissue biopsies were analysed [20,25,33,34]. Standard necropsy protocols generally covered other common cetacean viruses such as herpesvirus and morbillivirus [40,41], but it is only recently that recommendations have been developed for HPAI outbreaks in marine mammals [42]. Given the number of H5N1 cases reported in cetaceans is presently increasing, and the limitations of viral research in cetaceans, the risk posed by HPAI is herein investigated through protein sequence analysis. The screening of markers of adaptation of the avian virus to mammals reveals they are present in H5N1 clade 2.3.4.4b cetacean isolates, calling for an increase in surveillance. Phylogenetic analysis of co-occurring cases indicates that, to date, infection of cetaceans appears to be happening through spillover from seabirds and other marine mammals, with no evidence of a sustained transmission of the virus between cetaceans.

## 2. Materials and Methods

### 2.1. Review of IAV Cases Diagnosed in Cetaceans

Scientific literature was reviewed to identify cases of IAV infection in cetaceans using the National Center for Biotechnology Information NCBI PubMed database [43] as the main source of information. The searches had no date restriction and used MeSH search terms and Boolean operators (Influenza A, dolphins, whales, porpoises). Website links and scientific documents accessed from the bibliography of retrieved articles were also reviewed. A total of 34 articles, 13 scientific documents from the European Centre for Disease Prevention and Control (ECDC) [44,45,46,47,48,49,50,51,52,53,54,55,56], and 6 official websites [31,32,35,37,38,57] were reviewed. The information collected from each IAV infection case reported in a cetacean included the IAV subtype, the year and location of the case, the host species, the type of sample, the viral diagnosis method, and the disease symptoms. Figure 1, with the locations of reported cases of IAV H5N1 in cetaceans, was created in ArcGIS Pro 3.4.3 [58].

### 2.2. Analysis of Mammalian Adaptation Molecular Markers in Cetacean IAV Isolates

Viral amino acid sequences isolated from cetaceans were collected from the GISAID (global initiative on sharing all influenza data) EpiFlu^TM^ database [39] and from the NCBI Nucleotide database [59]. Sequences were downloaded in FASTA format for further analysis in the Molecular Evolutionary Genetic Analysis (MEGA) version 11 software [60]. For each IAV H5N1 isolate reported in cetaceans, amino acid sequences for seven essential viral proteins were downloaded (HA, NA, NP, PB2, PB1, and M1). A total of 211 sequences were acquired since 26 IAV H5N1 infection cases were reported for six species of cetaceans. Duplicated sequences or sequences belonging to the same sample were not included in the study, consequently 158 sequences from 21 isolates from cetacean infection cases were used. The reference sequence for amino acid numbering of H5N1 HA proteins was A/Vietnam/1203/2004 (H5N1) (GISAID EPI1990181), while the rest of H5N1 IAV proteins were numbered according to H5N1 genome segments of A/Goose/Guangdong/1/1996 (NCBI reference sequence NA: NC_007361.1; PB2: NC_007357.1; PB1: NC_007358.1; PA: NC_007359.1; NP: NC_007360.1; M1: NC_007363.1). A total of 40 molecular markers shortlisted by the European Food Safety Authority (EFSA) expert panel for preparedness, prevention, and control related to zoonotic avian influenza [22] were screened.

### 2.3. Phylogenetic Analysis of PB2 and HA Sequences in Cetacean and Marine Host Species

Viral amino acid sequences for PB2 and HA were obtained from GISAID EpiFlu^TM^ database by setting the search filters to details from cetacean isolates. The IAV “Type, H, N, and Clades” of the basic filters were set to H5N1 clade 2.3.4.4b, and the “Host” was set to animals and other mammals. In the additional filters, the “Location” was set to South America—Chile, South America—Peru, North America—United States—Florida, North America—Canada—Quebec, Europe—Sweden—Vastra Gotalands Lan, and Europe—United Kingdom and the “Collection Date” of the cetacean was set using a time frame of ±1 month regarding the cetacean isolate collection date (28 February 2022 to 30 April 2022 and 30 November 2023 to 31 January 2024 for North America—United States—Florida; 29 May 2022 to 29 July 2022 for Europe—Sweden—Vastra Gotalands; 5 August 2022 to 5 October 2022 for North America—Canada—Quebec; 22 October 2022 to 22 December 2022 for South America—Peru; 20 January 2023 to 6 April 2023 for Europe—United Kingdom; and 27 February 2023 to 1 May 2023 for South America—Chile). The host selection was restricted to species defined as marine in the World Register of Marine Species (WoRMS) [61]. Samples and sequences with undetermined amino acids were removed. Viral amino acid sequences for PB2 and HA obtained from marine animal IAV isolates, including cetaceans, and reported in the same regions and time period as that obtained from cetaceans, were aligned as described above. Aligned sequences were trimmed and maximum likelihood phylogenetic tree reconstruction was performed online in IQ-Tree using the FLU substitution model [62,63]. The tree for PB2 was inferred using a fragment of 759 amino acids with 92 informative sites in 96 sequences selected according to our established criteria. The tree for HA was inferred using a fragment of 567 amino acids with 71 informative sites in 99 sequences selected according to our established criteria. Bootstrap consensus trees from 1000 replicates were plotted using CHIPLOT 2.6.1 [64].

## 3. Results

### 3.1. The Number of Cases of IAV Infection Reported in Cetaceans Is Increasing

Our review of scientific literature indicates that to date LPAI IAV subtypes H1N3, H13N2, H13N9, and HPAI H5N1 have been isolated from cetacean species (Table 1). Since the recording of IAV cases in cetaceans, only ten cetacean species have been found to be susceptible to IAV infection. In approximately three decades (from 1975 to 2006) IAV was reported to infect species classified in the two suborders of cetaceans (*Mysticeti* and *Odontoceti*) and assigned to four families (*Balaenopteridae*, *Delphinidae*, *Monodontidae,* and *Phocoenidae*). These species were the long-finned pilot whale (*Globicephala melas*), the beluga whale (*Delphinapterus leucas*), the Dall’s porpoise (*Phocoenoides dalli*), and the common minke whale (*Balaenoptera acutorostrata*), and all were wild animals. No cases of IAV infection in cetaceans were documented in the scientific literature from 2007 to 2021. Since 2022, and, to date (October 2025), an increase in the number of IAV infection cases recorded in cetaceans is observed, and all cases are related to the HPAI subtype H5N1 of clade 2.3.4.4b [24,25,26,27,28,29,30,31,32,33,34,35,36,37,38,39]. These recent cases include six host species, all classified in the suborder *Odontoceti* and in families *Delphinidae* and *Phocoenidae* (Figure 1). There is no standardised process for the description of symptoms, diagnosis, or detection methods of the virus in the IAV infection cases reported in cetacean species. Concerning marine ecosystems, IAV infections have been documented in cetaceans inhabiting the Pacific Ocean [24,26]; the Pacific coast of South America [20,35,36]; both coasts of the Atlantic Ocean [30,33,37,38]; and the Arctic Ocean [27,28] to the North Sea [34].

### 3.2. IAV H5N1 Isolated from Cetaceans Have Mammal Adaptation Molecular Markers

A total of 40 molecular markers or mutations shortlisted by EFSA for preparedness, prevention, and control related to avian influenza adaptation to mammals [22] were inspected in H5N1 sequences isolated from cetacean hosts (Table 2). This analysis aimed to determine the existence of mutations that could reveal an adaptation of the virus to mammal hosts, and specifically to cetaceans. The mutations were inspected in the amino acid sequences of seven essential proteins including the surface proteins HA and NA, the proteins of the viral polymerase (PB2, PB1, PA) and ribonucleoprotein complex (NP), and the structural protein M1. Only six mutations were detected in 21 H5N1 sequences obtained from cetaceans. No molecular markers related to adaptation to mammals were found in viral amino acid sequences obtained from cetaceans for proteins NA, PB1, PA, or M1. Eight mutations were screened in the HA surface protein, of crucial importance in the binding of IAV to the host receptor, and only one mutation, HA:156A, was detected. For HA the reference sequence was A/Vietnam/1203/2004 that shows a K at amino acid 156. This marker of viral adaptation to mammals was observed in all the analysed sequences, which were isolated worldwide since 2022 from six cetacean species: Atlantic white-sided dolphin (*Lagenorhynchus acutus*), Bottlenose dolphin (*Tursiops truncatus*), common dolphin (*Delphinus delphis*), Chilean dolphin (*Cephalorhynchus eutropia*), Harbour porpoise (*Phocoena phocoena*), and Burmeister’s porpoise (*Phocoena spinipinnis*).

Viral proteins besides HA were aligned to reference sequences for subtype H5N1 obtained from the original case A/Goose/Guangdong/1/1996. In the PB2 subunit of the viral polymerase, essential for viral replication within the host cell, four mutations of the 15 markers screened in PB2 have been detected in sequences obtained since 2023 from cetaceans worldwide. Mutation A588V has only been reported in one of the most recent IAV isolates from dolphins of the USA Atlantic coast. Mutation Q591K was detected in isolates of 2023 in Chile from two Burmeister’s porpoises. Mutation E627K was reported in isolates from Bottlenose dolphins of the USA, specifically from 2023 to 2024 in Florida. Mutation D701N was detected in all the isolates from Burmeister’s porpoises and Chilean dolphins, all reported in Chile in 2023, including several cases harbouring mutation Q591K.

In relation to markers screened in the NP, mutation Y52H is present in a single sequence isolated from a Harbour porpoise in Sweden in the North Sea in 2022. Two molecular markers were not detected in viral amino acid sequences isolated from cetaceans (Appendix A) but did show a mutation in relation to the reference sequences. Mutation HA:208T was not detected although all sequences obtained from cetaceans had a K differing from the reference that harboured a Q. In the case of mutation PA:85I, several recent cases showed amino acid A instead of the T in the reference sequence. The molecular markers listed by EFSA and related to IAV adaptation to mammals screened that were not observed in viral amino acid sequences obtained from cetacean IAV isolates were 156V, 186V, 186D, 208T, 221D, 222L, and 224S in HA; 399R and 432E in NA; 9N, 199S, 271A, 292V, 526R, 588I, 591R, 627V, 631L, 702R, and 740N in PB2; 66S in PB1; 85I, 97I, 186S, 336M, 356R, and 552S in PA; 52N, 100I, 100V, 283P, 313V, and 313Y in NP; and 95K in M1 (Appendix A).

### 3.3. Phylogenetic Analysis of PB2 and HA Supports Cross-Species Transmission of IAV H5N1 to Cetaceans

HPAI H5N1 could be circulating within certain cetacean species, or it could be reaching cetaceans through cross-species transmission from other host species (cetaceans, pinnipeds, or seabirds). We addressed this question through phylogenetic analysis of the PB2 subunit of the viral polymerase and the HA that binds the host cell receptor, as these two proteins harbour the higher number of mutations related to adaptation to mammals (Table 2). PB2 and HA amino acid sequences were downloaded from the EpiFlu^TM^ GISAID database filtering data by subtype H5N1 clade 2.3.4.4b, the location, and collection date of the cetacean isolates, and retrieving only sequences isolated from marine animals. The consensus bootstrap maximum likelihood tree for PB2 (Figure 2, left) and HA (Figure 2, right) show cetacean isolates cluster with statistical support with other isolates from marine animals from the same marine ecosystems. Sequences isolated in 2022 and 2023 from cetaceans inhabiting the Celtic Sea and the North Sea of the North-East Atlantic Ocean (NEAO, coloured in light blue in Figure 2) include isolates from four cetaceans, two dolphins (depicted as stars) and two porpoises (depicted as triangles), and cluster with seabird isolates. Another cluster of sequences obtained also in 2022 and 2023 from marine animal hosts relates to the South-East Pacific Ocean (SEPO, coloured in dark blue in Figure 2), including isolates from six cetaceans, three porpoises and three dolphins, that cluster with sequences obtained from pinnipeds and seabirds. PB2 and HA sequences from 2022 to 2024 isolates obtained from marine animals inhabiting the North-West Atlantic Ocean (NWAO, coloured in blue in Figure 2), from Canada and the USA, group in several branches. Eight dolphin isolates significantly cluster with seabirds in the case of two isolates from 2022, or other dolphins in the case of NWAO dolphin isolates from 2023 to 2024 (five sequences for PB2 and six for HA, as the PB2 sequence of isolates with ID 20055121 was removed during sequence alignment). A single PB2 sequence obtained in a Bottlenose dolphin in 2025 included in the PB2 tree (Figure 2, left) also clusters with isolates from 2023 to 2024. The HA consensus tree (Figure 2, right) includes up to three dolphin sequences from 2025 that outgroup to NEAO and NWAO. These isolates from 2025 are from an unknown location in USA, however given they cluster with sequences from marine animals of the Atlantic Ocean, they have been coloured as NWAO. Isolate EPI_ISL_19825688 from a dolphin in USA 2025 has no registered sequence for PB2 in the GISAID EpiFlu^TM^ database. Altogether, consensus trees inferred from PB2 and HA sequences strongly support cross-species transmission of H5N1 to cetaceans, rather than sustained circulation of the virus within cetacean species.

## 4. Discussion

Virology research in marine mammals is challenged by several limitations including difficulties in sampling and scarce diagnostic protocols. This study focuses exclusively on IAV infection in cetaceans. No mass mortality events in cetaceans have so far been related to IAV. Massive die-off events in cetaceans have been associated with other viral infectious diseases. An example is the death of hundreds of striped dolphins caused by a morbillivirus in the Mediterranean Sea [65]. IAV has caused mass mortality events in other marine mammals, for instance the death of hundreds of Harbour seals infected with the subtype H7N7 [66]. No IAV cases have been described in other *Delphinoidea* species such as beaker (*Globicephala* sp.) or killer whales (*Orcinus* sp.). No data were documented between 2006 and 2021, with only two cetacean families, *Phocoenidae* and *Delphinidae*, having cases before and after this gap or an apparent cetacean IAV-free period. This observation is important, as it points to these marine mammals as not highly susceptible to cross-species transmission from avian species. Since the appearance of the goose/Guangdong (gs/Gd) lineage of IAV, the most successful clade was clade 2, from which different strains evolved into subtypes that spread widely. Between 2007 and 2021 subtypes like H5N2, H5N5, or H5N6 predominated worldwide, but just before the appearance of H5N1 there were only occasional outbreaks. Finally, in 2021 the HPAI subtype H5N1 clade 2.3.4.4b appeared and prevailed until today [67]. A year later the spillover to cetaceans started but never caused massive mortal events as in other marine mammals such as pinnipeds [15,20,68].

The number of genome fragments of IAV cetacean isolates in public databases is increasing thanks to advances in sequencing technologies. These data are crucial for the scientific community for prevention, preparedness, and control of future IAV epizootic outbreaks in marine animals. EFSA experts classified the shortlist of mutations to screen in five phenotypic trait groups related to adaptation to mammals [22]. H5N1 clade 2.3.4.4.b IAV amino acid sequences observed in cetacean isolates include mutations classified in three of the five phenotypic trait groups proposed [22]. All cetacean isolates include the marker 156A in HA, classified in the trait that increases mammalian specificity of virus attachment to receptor [22]. The structure of the sialic acids in the host cell, and particularly their glycosidic bonds, influences cross-species transmission of IAV [69]. Mutation T156A (position 160 when numbered in H3) is known to increase virus binding to α2–6 sialic acids and remove a glycosylation site in positions 154–156 of H5 HA, and facilitates virus transmission in mammals [70,71]. The two most common forms of sialic acid in higher vertebrate animals are N-acetylneuraminic acid (Neu5Ac) and N-glycolylneuraminic acid (Neu5Gc) [69], and whale nasal cartilage was found to contain only Neu5Ac in an early study [72], although further characterisation of sialic acids in cetaceans is needed. Lack of attachment to tracheal and bronchial epithelium of Harbour porpoises and Bottlenose dolphins has been reported for IAV subtype H4N5 and H7N7 [73], in contrast to attachment to the respiratory tract of Harbour seals, in consistency with outbreaks in pinnipeds. The mutations in PB2 are classified in a trait group related to increased activity of the viral polymerases in mammalian hosts (588V, 591K, 627K, 701N). The optimal function of IAV viral polymerase depends on species-specific differences in the acidic nuclear phosphoprotein 32 (ANP32) family of proteins [74]. Substitutions that adapt the viral polymerase to shorter mammalian ANP32 are selected in HPAI infecting mammalian hosts [75]. In mammals, these proteins need a positively charged groove to bind and stabilise the viral polymerase [76]. Therefore, mutations such as 591K or 627K that induce a change to positively charged amino acids lead to a better attachment of PB2 to ANP32 proteins in mammals. 627K, an important and well-known mutation in PB2, was detected only in five of the six Bottlenose dolphin isolates reported in USA in 2023 and 2024, while 591K was only present in two Burmeister’s porpoises from Chile. Mutation 701N in PB2 is a mammalian marker in avian samples [67] and it is present in all the sequences of cetaceans from Chile in the SEPO, where the most lethal IAV outbreaks in marine mammals were reported [15,19,20,76], suggesting cross-species transmission of H5N1 between marine mammals [15,19,77]. The combination E627K/D701N/S714R was described as an enhancer of the IAV polymerase, increasing its virulence and replication [78]. Cetacean isolates only lack S714R, a mutation that should be surveyed closely in all amino acid sequence data reported from cetacean hosts. Another important mutation in PB2 that increases the activity of the viral polymerase is T271A, even though its effect is not as important as E627K [75]. Mutation T271A, although not included in the EFSA molecular markers screened in this study, was not detected in the viral sequences from cetaceans (Appendix A). The only marker observed in NP (52H) is classified in a trait group related to evasion of innate immunity and counteraction of mammalian restriction factors. This mutation is related to altering the antiviral activity of BTN3A3 (butyrophilin subfamily 3 member A3) that evades avian viruses inhibiting IAV RNA replication through a mechanism that involves residue 313 adjacent to 52 in the NP structure [79]. Other phenotypic trait marker groups related to adaptation to mammals that were not observed in cetacean isolates are related to increased HA stability in the mammalian environment, and disruption of second sialic acid binding site in NA, as named in [22]. An adequate evaluation of the risk of IAV adaptation to cetacean species requires future research concerning species-specific virus–host cellular and molecular mechanisms.

The most accepted hypothesis for the cross-species transmission of IAV to cetaceans is the avian origin, and also for marine animals in general, and it is supported by phylogenetic analysis presented in Figure 2. Previous articles have reached the same conclusion, like [29], that determines which was the first infection of cetaceans from a gull-origin IAV; Ref. [80], where phylogenetic analyses were also performed; or ref. [34], that suggests spillover from wild birds as the most probable explanation. Reducing H5N1 outbreaks in wild birds through vaccination of poultry has been proposed as a strategy to prevent global viral dissemination [6,81]. Phylogenetic analysis of PB2 and HA sequences (Figure 2) strongly support cross-species transmission of H5N1 to cetaceans, rather than sustained circulation of the virus within cetacean species. The inferred trees agree with the NEAO origin of the cross-species transmission of HPAI to cetaceans, from where the avian virus crossed the Atlantic Ocean and reached North America. From there, HPAI continued expanding across the American continent through the Gulf of Mexico and extending to the SEPO (Figure 1). Seasonal and endemic circulation of HAPI in aquatic birds of marine ecosystems is consistent with the fact that IAV cetacean isolates from different years cluster together in the same marine oceanic system (Figure 2). Several USA dolphin isolates in 2023 and 2024 even cluster together in NWAO. However, phylogenetic analysis does not support onward transmission of HPAI H5N1 among cetaceans, and mammalians appear to remain as what has been previously described as “dead-end” infections (2). In conclusion, surveilling IAV in cetaceans and other marine mammals is important, not just for their conservation, but also for their condition as sentinel species: they work as an early warning of environmental changes and potential threats for human health and other animals in their ecosystems [23]. The areas represented in Figure 1 show that there are regions of the planet with no feedback regarding IAV H5N1 infections in cetacean species. Considering their extensive coastlines, it is noteworthy that many countries in Africa, Asia, and Oceania have not reported IAV isolates from cetaceans. A lack of surveillance of HPAI in cetaceans would ignore the host range extension of H5N1. Nevertheless, it is extremely unlikely that all sick or dead animals were sampled and reported, especially in remote areas where observations and surveillance efforts are scarce [82]. To deal with HPAI and other panzootics affecting marine animals as a global issue, marine policy programmes should promote surveillance of IAV in cetacean wild populations worldwide and facilitate resources for surveillance in developing country regions. Diagnosis of IAV should be a routine protocol when samples are taken from bycaught or stranded cetaceans. This will allow a better evaluation of the consequences of IAV, not only in cetaceans, but in wildlife populations, but also a better understanding of IAV evolution, leading to a more efficient control of animal and human health [82]. The inclusion of marine ecosystem and marine animal health is thus essential in the One Health approach to panzootics and pandemics.

## Figures and Tables

**Figure 1 viruses-17-01536-f001:**
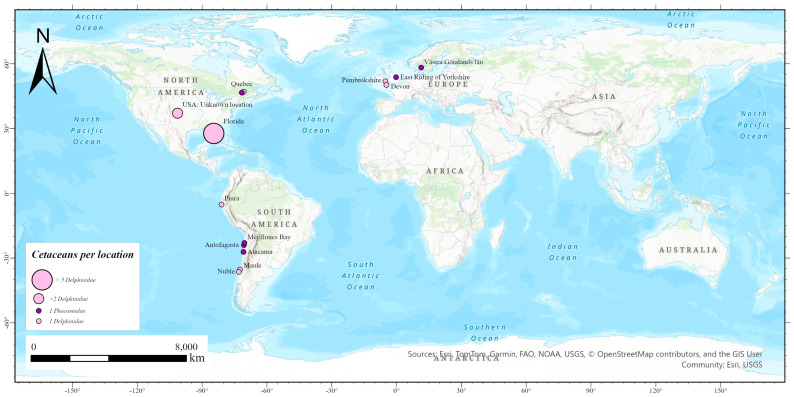
Location of reported cases of IAV H5N1 (clade 2.3.4.4b) worldwide in cetaceans since the outbreak of 2021 that have provided the viral sequences used in the study. Cases were obtained from GISAID EpiFlu^TM^ database and the bibliography. The sequences obtained from dolphin samples are depicted as pink dots of different sizes according to the number of cases reported in the same area. This number varies from 1 (in Nuble, Maule, Piura, Quebec, Pembrokshire and Devon) to 5 (in Florida). Only 3 cases could not be located in a specific region and have been placed inland (“USA: Unknown location”). The sequences from porpoise samples are depicted as purple dots of one size because only one specimen has been reported in each location (Atacama, Antofagasta, Mejillones Bay, Quebec, East Riding of Yorkshire and Västra Götalands län). Map created in ArcGIS Pro 3.4.3.

**Figure 2 viruses-17-01536-f002:**
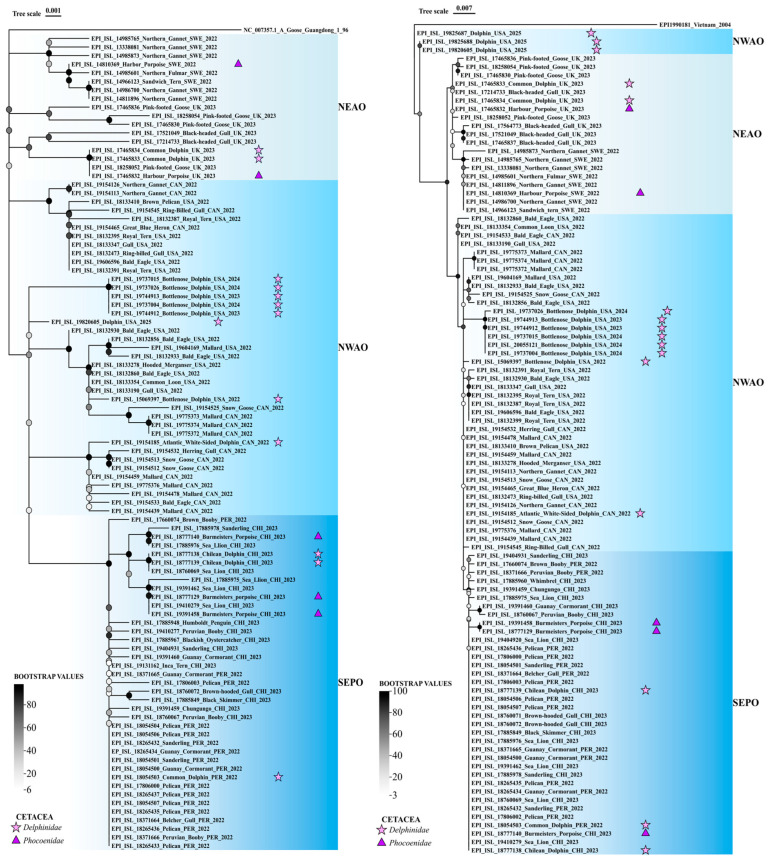
PB2 (**left**) and HA (**right**) phylogenetic trees inferred from cetacean and co-occurring marine animal H5N1 isolates. PB2 amino acid sequences were obtained from the GISAID EpiFlu^TM^ database filtering IAV H5N1 (clade 2.3.4.4b), marine animal hosts, and location and time of collection of cetacean isolates. The bootstrap consensus tree from 100 replicates is illustrated. A fragment of 759 amino acids with 92 informative sites in 96 sequences was used for PB2, and for HA, a fragment of 567 amino acids with 71 informative sites in 99 sequences was analysed. Cases of cetaceans are depicted with stars (dolphins) or triangles (porpoises). Oceanic marine ecosystem collection locations are coloured in blue shades and are North-East Atlantic Ocean (NEAO), South-East Pacific Ocean (SEPO), and North-West Atlantic Ocean (NWAO). Isolate EPI_ISL19825688 from a dolphin in USA 2025 has no registered sequence for PB2 in GISAID EpiFlu^TM^ database. USA isolates from 2025 are from unknown locations, however given they cluster with sequences from marine animals of the Atlantic Ocean, they have been coloured as NWAO.

**Table 1 viruses-17-01536-t001:** Summary of documented cases of IAV infection in cetaceans (cases reported until 31 October 2025).

Year	Subtype	Host	Family	Species	Location	Sample ^d^	Reference
1975/76	H1N3	Common minke whale ^b^	*Balaenopteridae*	*---*	South Pacific	Lung and liver	[24]
1984	H13N2 and H13N9	1 Long-finned pilot whale	*Delphinidae*	*Globicephala melas*	Coast of New England	Lung and hilar node	[25,29]
1990–1991	HXNX	5 Beluga whales	*Monodontidae*	*Delphinapterus leucas*	Southeast Baffin Island (Canada)	Serum	[28]
2000–2001	HXNX ^a^	2 Dall’s porpoises	*Phocoenidae*	*Phocoenoides dalli*	Western north Pacific	Serum	[26]
	HXNX ^a^	7 Common minke whales	*Balaenopteridae*	*Balaenoptera acutorostrata*	Western north Pacific	n.i.	
2001–2006	HXNX	Beluga whales ^b^	*Monodontidae*	*Delphinapterus leucas*	Van Mijenfjorden, Svalbard (Norway)	Blood	[27]
2022	H5N1	1 Bottlenose dolphin ^c^	*Delphinidae*	*Tursiops truncatus*	Dixie County (Florida, USA)	Ocular conjunctival, rectal, spiracle swabs	[30,31,32,33]
	H5N1	1 Harbour porpoise	*Phocoenidae*	*Phocoena phocoena*	Quebec (Canada)	Lung, bronchus, brain, kidney, liver, spleen, intestine, muscle and blubber	[30,31]
	H5N1	1 Atlantic white-sided dolphin	*Delphinidae*	*Lagenorhynchus acutus*	Quebec (Canada)		
	H5N1	1 Common dolphin	*Delphinidae*	*Delphinus delphis*	Piura (Peru)	n.i.	[20]
	H5N1	1 Harbour porpoise	*Delphinidae*	*Phocoena phocoena*	West Coast of Sweden	n.i.	[34]
2023	H5N1	2 Chilean dolphins	*Delphinidae*	*Cephalorhynchus eutropia*	Maule y Nuble (Chile)	n.i.	[35]
	H5N1	2 Burmeister’s porpoises	*Phocoenidae*	*Phocoena spinipinnis*	Antofagasta and Atacama (Chile)	n.i.	
	H5N1	1 Burmeister’s porpoise	*Phocoenidae*	*Phocoena spinipinnis*	Mejillones Bay (Chile)	Serum	[36]
	H5N1	2 Common dolphin	*Delphinidae*	*Delphinus delphis*	Devon (England) and Pembrokeshire (Wales)	n.i.	[37]
	H5N1	1 Harbour porpoise	*Phocoenidae*	*Phocoena phocoena*	East Riding of Yorkshire (England)	n.i.	
	H5N1	2 Bottlenose dolphins	*Delphinidae*	*Tursiops truncatus*	Florida (USA)	n.i.	[38,39]
2024	H5N1	4 Bottlenose dolphins	*Delphinidae*	*Tursiops truncatus*	Florida (USA)	n.i.	
2025	H5N1	3 Bottlenose dolphins	*Delphinidae*	*Tursiops truncatus*	Florida (USA)	n.i.	[39]

^a^: Subtype not possible to identify through immunodiagnostic methods using purified human H3N2 strain A/Aichi/2/68 as antigen. ^b^: Unknown number of positive cases. ^c^: Found dead. ^d^: n.i: no information available.

**Table 2 viruses-17-01536-t002:** Molecular markers of mammal adaptation detected in IAV H5N1 cetacean isolates.

	Viral Protein					
	HA ^b^	PB2				NP
IAV H5N1 clade 2.4.4.4b cetacean isolates ^a^	156A	588V	591K	627K	701N	52H
A/Goose/Guangdong/1/96		A	Q	E	D	Y
A/Vietnam/1203/2004	K					
EPI_ISL_13338081_Northern_Gannet_SWE_2022	A	A	Q	E	D	H
EPI_ISL_14810369_Harbour_Porpoise_SWE_2022	A	-	-	-	-	H
EPI_ISL_15069397_Bottlenose_Dolphin_USA_2022	A	-	-	-	-	-
EPI_ISL_19154185_Atlantic_white-sided_Dolphin_CAN_2022	A	-	-	-	-	-
EPI_ISL_18054503_Common_Dolphin_PER_2022	A	-	-	-	-	-
EPI_ISL_17465832_Harbour_Porpoise_UK_2023	A	-	-	-	-	-
EPI_ISL_17465833_Common_Dolphin_UK_2023	A	-	-	-	-	-
EPI_ISL_17465834_Common_Dolphin_UK_2023	A	-	-	-	-	-
EPI_ISL_18777140_Burmeisters_Porpoise_CHI_2023	A	-	-	-	N	-
EPI_ISL_18777139_Chilean_Dolphin_CHI_2023	A	-	-	-	N	-
EPI_ISL_18777129_ Burmeisters_Porpoise _CHI_2023	A	-	K	-	N	-
EPI_ISL_19391458_Burmeisters_Porpoise_CHI_2023	A	-	K	-	N	-
EPI_ISL_18777138_Chilean_Dolphin_CHI_2023	A	-	-	-	N	-
EPI_ISL_19744912_Bottlenose_Dolphin_USA_2023	A	-	-	K	-	-
EPI_ISL_19744913_Bottlenose_Dolphin_USA_2023	A	-	-	K	-	-
EPI_ISL_20055121_Bottlenose_Dolphin_USA_2024	A	-	-	-	-	-
EPI_ISL_19737004_Bottlenose_Dolphin_USA_2024	A	-	-	K	-	-
EPI_ISL_19737015_Bottlenose_Dolphin_USA_2024	A	-	-	K	-	-
EPI_ISL_19737026_Bottlenose_Dolphin_USA_2024	A	-	-	K	-	-
EPI_ISL_19820605_Dolphin_USA_2025	A	-	-	-	-	-
EPI_ISL_19825687_Dolphin_USA_2025	A	V	-	-	-	-
EPI_ISL_19825688_Dolphin_USA_2025	A	ns ^c^	ns ^c^	ns ^c^	ns ^c^	-

^a^: Reference amino acid sequences are A/Vietnam/1203/2004 (GISAID EPI1990181) for HA, and A/Goose/Guangdong/1/96 for PB2 (NCBI NC_007357.1) and NP (NCBI NC_007360.1). A recent avian isolate A/Northern gannet/Sweden/SVA220525SZ0402/FB001671/O-2022 was also included for HA (GISAID EPI2069014), PB2 (GISAID EPI2069011), and NP (GISAID EPI2069015). Cetacean isolates are named with GISAID EpiFlu^TM^ identification number. USA: United States of America; SWE: Sweden; CAN: Canada; UK: United Kingdom; CHI: Chipre; PER: Peru. ^b^: No markers of mammal adaptation have been detected in the cleavage site motif of HA. ^c^: No sequence retrieved (ns). -: The amino acid in the cetacean IAV isolate is identical in the reference sequence.

## Data Availability

The original contributions presented in this study are included in the article with detailed information on the public domain sequence data used for analysis.

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
