# Peer review of "Concern for Highly Pathogenic Avian Influenza Spillover into Cetaceans"

_viruses, 2025, doi:10.3390/v17121536_

Round 1

Reviewer 1 Report

Comments and Suggestions for Authors

Pérez-Sánchez, Báez and Johnstone present this manuscript titled “Concern for highly pathogenic avian influenza spillover into cetaceans”. The manuscript is well written, informative and interesting for the wider readership as this is an important topic.

However I have significant concerns on the organisation of the manuscript, as it presents as a literature review or discussion/opinion of the topic, with the authors actual research embedded within the cited literature.

Introduction and discussion material is located in every section of the manuscript, the authors have not followed the clear introduction/results/discussion format for scientific articles, which makes it very difficult to review the actual work carried out by the authors.

Please restructure the manuscript so that all cited introductory material is in the introduction section.

Please restructure the results section so that it only lists the research carried out by the authors for this paper.

Please restructure the discussion to only discuss the results presented by the authors in this paper only, with the context of the introduction.

The introduction reads more like a textbook introduction to influenza virus – this level of detail is not necessary as it can be cited. Please briefly summarise the gene segments related to the work carried out in the manuscript, and cite the relevant papers on the full characterisation.

Key to this article is the context needed by the readers of this manuscript for work with cetaceans. How reliable are detection methods, how much surveillance happens worldwide on these species? Some of this information is in the discussion and should be brought forward to the introduction.

After restructuring the manuscript, it will be more clear what work is actually being proposed in this MS.  

As work on cetaceans seems limited at best, the authors should also include the limitations of this work .

Line 32, specified spherical virus, there is much evidence on both spherical and filamentous particles in IAV in multiple contexts, suggest to remove “spherical”.

Line 51-52, the authors should use the word “subtype” in their description of HA/NA classification of IAVs

Line 71, instead of “four years ago”, please say the year and cite with appropriate literature for this “great breakthrough”, and please describe what the authors mean by this term?

Line 88-89, please could the authors provide some context on the surveillance efforts in cetaceans? Is AIV testing regularly carried out on dead animals, or only targeted efforts?

Line 93, please cite the paper with the phylogenetic analysis of H5, showing lack of samples/evolution in cetaceans.

E.g. lines 175-180. These are cited results from other papers, and should be in the introduction to allow the authors to use this literature as context for their study.

Table 2, while useful to have GS/GD and clade 1 H5N1 as the references, it would be useful to include a more recent strain (e.g. precursors of mammalian infections) included. e.g gyrfalcon/14 or astrakhan/20

Please describe in further detail the adaptive mutations the authors suggest are specific to cetaceans, and any supporting evidence in the published literature.

line 221, the authors suggest A/GS/GD is clade 2.3.4.4b

Figure 1, seemingly GS/GS and VN/1204 are coming out more evolved in this phylogeny, based on branch length – please could the authors explain this?

Introduction in the discussion, e.g. 301 to 302, 314-316

Figure 2 is in the discussion, this should be introductory material

Author Response

Response to Reviewer 1

Thank you for your comments and time dedicated to review our manuscript. We have introduced major changes including reorganization of main sections of the manuscript. Also, we have improved figure quality and English. Please find the detailed responses below and the corresponding revisions in track changes in the re-submitted files.

Comments of Reviewer 1: Pérez-Sánchez, Báez and Johnstone present this manuscript titled “Concern for highly pathogenic avian influenza spillover into cetaceans”. The manuscript is well written, informative and interesting for the wider readership as this is an important topic.

However I have significant concerns on the organisation of the manuscript, as it presents as a literature review or discussion/opinion of the topic, with the authors actual research embedded within the cited literature.

Introduction and discussion material is located in every section of the manuscript, the authors have not followed the clear introduction/results/discussion format for scientific articles, which makes it very difficult to review the actual work carried out by the authors.

Please restructure the manuscript so that all cited introductory material is in the introduction section.

Please restructure the results section so that it only lists the research carried out by the authors for this paper.

Please restructure the discussion to only discuss the results presented by the authors in this paper only, with the context of the introduction.

Response: Thank you for your comments. Concerning the organisation of the manuscript, we have restructured introduction, results and discussion accordingly. Table 1 has been moved to the Introduction. Part of the sentences previously included in section 3.1 of Results explaining in detail references compiled in Table 1 have been moved to the last paragraph of the Introduction. Section 3.1 is now exclusively dedicated to refer to the conclusions after compiling all the reviewed literature in regard to the IAV subtype, the cetacean species infected by IAV, and the marine ecosystem (location), and scarce information on description of symptoms and diagnosis. Several sentences included in the discussion section are now in the introduction or in the results section, as detailed in specific comments bellow.

Comments of Reviewer 1: The introduction reads more like a textbook introduction to influenza virus – this level of detail is not necessary as it can be cited. Please briefly summarise the gene segments related to the work carried out in the manuscript, and cite the relevant papers on the full characterisation.

Response: Thank you for your suggestion. The basic knowledge of influenza virus has been summarised, and paragraphs 1 and 2 of the introduction have been merged in a single paragraph (page 1, lines 34-42). Only the seven essential proteins analysed in the results section are mentioned (HA, NA, PB2, PB1, PA, NP and M1).

Comments of Reviewer 1: Key to this article is the context needed by the readers of this manuscript for work with cetaceans. How reliable are detection methods, how much surveillance happens worldwide on these species? Some of this information is in the discussion and should be brought forward to the introduction.

Response: Thank you for this observation. In relation to surveillance in cetaceans a sentence previously included in the discussion has been moved and extended in the introduction, citing a new reference. Specifically the sentence reads as “Standard necropsy protocols generally covered other common cetacean viruses such as herpesvirus and morbillivirus [40, 41], but it is only recently that recommendations have been developed for HPAI outbreaks in marine mammals [42]” (page 3, lines 98-100). In relation to this matter, we have also included a sentence in the abstract “Sampling difficulties and limited tests available for diagnosis are a challenge to cetacean virology research” (lines 14-15).

Comments of Reviewer 1: After restructuring the manuscript, it will be more clear what work is actually being proposed in this MS. As work on cetaceans seems limited at best, the authors should also include the limitations of this work .

Response: Thank you very much for pointing this out. We have made clear what work is actually being proposed and the limitations of IAV research in cetaceans. Besides in the abstract as mentioned above, in the final sentences of the Introduction, we have edited a sentence that now reads “Given the number of H5N1 cases reported in cetaceans is presently increasing, and the limitations of viral research in cetaceans, the risk posed by HPAI is herein investigated through protein sequence analysis” (page 3, lines 101-103).

Comments of Reviewer 1: Line 32, specified spherical virus, there is much evidence on both spherical and filamentous particles in IAV in multiple contexts, suggest to remove “spherical”.

Response: Thank you, the word “spherical” has been deleted from the sentence in line 35 (page 1).

Comments of Reviewer 1: Line 51-52, the authors should use the word “subtype” in their description of HA/NA classification of IAVs

Response: Accordingly, the word “subtype” has been used throughout the manuscript when referring to HA/NA IAV classification.

Comments of Reviewer 1: Line 71, instead of “four years ago”, please say the year and cite with appropriate literature for this “great breakthrough”, and please describe what the authors mean by this term?

Response: The sentence has been edited substituting “great breakthrough” by “expansion” and “four years ago” to “in 2021”. The complete sentence reads “Even though its expansion happened in 2021, its presence has been detected in a huge range of mammal species since 2003, generating concern as its adaptation to mammals implies an important threat for wild mammal species and human health [14]” (page 2, lines 62-65).

Comments of Reviewer 1: Line 88-89, please could the authors provide some context on the surveillance efforts in cetaceans? Is AIV testing regularly carried out on dead animals, or only targeted efforts?

Response: Thank you for pointing this out, surveillance efforts in cetaceans on IAV are scarce in contrast to other viruses. This has been mentioned in the introduction in the sentence commented before that reads “Standard necropsy protocols generally covered other common cetacean viruses such as herpesvirus and morbillivirus [40, 41], but it is only recently that recommendations have been developed for HPAI outbreaks in marine mammals [42]” (page 3, lines 98-100).

Comments of Reviewer 1: Line 93, please cite the paper with the phylogenetic analysis of H5, showing lack of samples/evolution in cetaceans.

Response: The end of the final sentence of the introduction written “as to date, phylogenetic analysis do not support a sustained transmission of the virus between cetaceans”, aimed to summarize the phylogenetic analysis of amino acid sequences obtained from cetacean isolates. The sentences written to end the introduction now read “Phylogenetic analysis of co-occurring cases indicates that, to date, infection of cetaceans appears to be happening through spillover from sea-birds and other marine mammals, with no evidence of a sustained transmission of the virus between cetaceans” (page 3, lines 105-108).

Comments of Reviewer 1: E.g. lines 175-180. These are cited results from other papers, and should be in the introduction to allow the authors to use this literature as context for their study.

Response: Thank you, as explained above, the cited results from other papers have been moved to the fourth and final paragraph of the introduction.

Comments of Reviewer 1: Table 2, while useful to have GS/GD and clade 1 H5N1 as the references, it would be useful to include a more recent strain (e.g. precursors of mammalian infections) included. e.g gyrfalcon/14 or astrakhan/20

Response: Thank you for the suggestion. Reference sequences gyrfalcon/14 or astrakhan/20 are not H5N1 and thus we have decided to include instead another H5N1 reference sequence in table 2. Specifically a 2022 isolate from a Northern Gannet from Sweden that we had already included in the phylogenetic analysis.

Comments of Reviewer 1: Please describe in further detail the adaptive mutations the authors suggest are specific to cetaceans, and any supporting evidence in the published literature.

Response: Throughout the manuscript mutations that are specific to mammals have been discussed, but no mutations have been suggested to be exclusive or specific to cetaceans.

Comments of Reviewer 1: line 221, the authors suggest A/GS/GD is clade 2.3.4.4b

Thank you for pointing this out, clade 2.3.4.4b has been deleted from the sentence now placed in page 8 line 223. Accordingly, in the introduction, clade 2.3.4.4b has also been deleted from the sentence in page 2 line 57 in the re-submitted manuscript.

Comments of Reviewer 1: Figure 1, seemingly GS/GS and VN/1204 are coming out more evolved in this phylogeny, based on branch length – please could the authors explain this?

Response: The maximum likelihood phylogenetic analysis infers a common ancestor to all sequences analysed. A longer branch is separating sequences that have evolved differently from a common ancestor given it implies more differences or mutations between branches or nodes. Because evolution is not directional, a longer branch does not imply more evolution or more time, it means more changes or mutations between different sequences in relation to a common ancestor sequence.

Comments of Reviewer 1: Introduction in the discussion, e.g. 301 to 302, 314-316

Response: Thank you for your suggestion. The referred sentence is discussing the fact that there is no information on IAV in certain species of the Delphinoidea taxonomic category, and has not been moved to the introduction.

Comments of Reviewer 1: Figure 2 is in the discussion, this should be introductory material

Response: Thank you for your suggestion, we have commented the figure in the discussion and moved the figure to the results section. Figure 2 is now re-named Figure 1 as it shows the location of IAV H5N1 cases in cetacean species and the type of cetacean (dolphin or porpoise).

Reviewer 2 Report

Comments and Suggestions for Authors

In this manuscript, the authors characterized the the avian influenza viruses infection in cetaceans, and found the H5N1 viruses spillover from seabirds have led to increasing reports in cetaceans. This study contributed our further understanding on the global circulated H5N1 viruses in mammals.

  1. In the introduction section, there too many words to described the basic knowledge of influenza virus (paragraph 1 and 2). A brief introduction of influenza virus is recommended.
  2. A logical introduction of H5N1 viruses in migratory birds, domestic poultry, mammals including dairy cows and humans is essential to improve the quality of the manuscript.
  3. It is recommended to use a separate paragraph to introduce reports on the infection of cetaceans by low-pathogenic avian influenza viruses (H1, H13).
  4. The cleavage site motif of HA should be listed in Table 2.
  5. More molecular makers of mammalian adaptation of H5N1 viruses should be analyzed, such as position 30 and 215 in M1 protein (PMID: 34908445).
  6. The phylogenetic trees of HA and PB2 should be improved to reveal the evolutionary dynamics of H5N1 viruses in mammals.
  7. Although the map with the locations of H5N1 viruses in cetaceans is provided in figure 2, the global Migratory Flyway in figure 2 is recommended to support the relationships of migratory seabirds and cetaceans infections. Thedistribution of avian influenza viruses has close relationship with the migratory Anseriformes and Charadriiformes (https://doi.org/10.1016/j.jia.2023.12.033).
  8. In the discussion section, there are large numbers of words to discuss the H5N1 viruses, however, there no word to discuss prevention of H5N1 by vaccine. Vaccination of highly pathogenic H5 viruses in domestic have been proven the most effective way to prevent global circulation of H5 (PMID: 34757542). Discussion of vaccination of H5N1 viruses is recommended in discussion.

Author Response

Response to Reviewer 2 Comments

Thank you for your comments and time dedicated to review our manuscript. We have introduced major changes including reorganization of main sections of the manuscript. Also, we have improved figure quality and English. Please find the detailed responses below and the corresponding revisions in track changes in the re-submitted files.

Comments of Reviewer 2: In this manuscript, the authors characterized the the avian influenza viruses infection in cetaceans, and found the H5N1 viruses spillover from seabirds have led to increasing reports in cetaceans. This study contributed our further understanding on the global circulated H5N1 viruses in mammals.

In the introduction section, there too many words to described the basic knowledge of influenza virus (paragraph 1 and 2). A brief introduction of influenza virus is recommended.

Response: Thank you, following your suggestion the basic knowledge of influenza virus has been summarised, and paragraphs 1 and 2 of the introduction have been merged in a single paragraph (page 1, lines 34-42). Only the seven essential proteins analysed in the results section are mentioned (HA, NA, PB2, PB1, PA, NP and M1).

Comments of Reviewer 2: A logical introduction of H5N1 viruses in migratory birds, domestic poultry, mammals including dairy cows and humans is essential to improve the quality of the manuscript.

Response: Thank you for pointing this out. We have introduced a new sentence and new citations in the introduction, line “This mammal-to-mammal transmission has been reported for domestic animals such as minks in Spain [16] or dairy cows in the USA [17], and leading to sporadic cases in humans [18].” (page 2, line 68-70).

Comments of Reviewer 2:

It is recommended to use a separate paragraph to introduce reports on the infection of cetaceans by low-pathogenic avian influenza viruses (H1, H13).

Response: Thank you for your suggestion. We have specified LPAI and HPAI when commenting on IAV infections in cetaceans in the introduction. However, we have not separated LPAI information in a separate paragraph in order to include general comments on diagnostic protocols and symptoms for both LPAI and HPAI. Also, Table 1 has been moved to the introduction and we believe it demands a single paragraph with no partitions.

Comments of Reviewer 2: The cleavage site motif of HA should be listed in Table 2.

Response: The cleavage site motif of HA does not include any mutations in the sequences isolated from cetaceans, which has been detailed in the footnote of Table 2.

Comments of Reviewer 2: More molecular makers of mammalian adaptation of H5N1 viruses should be analysed, such as position 30 and 215 in M1 protein (PMID: 34908445).

Response: Thank you. The suggested markers have been screened and they are not present in M1 protein sequences obtained from cetacean hosts. We have not included this information to avoid a skew to certain mutations, as there are many other mutations that could be screened. In fact, this is the reason why we decided to follow EFSA recommendations after reading extensive literature concerning IAV molecular markers of adaptation to different hosts.

Comments of Reviewer 2: The phylogenetic trees of HA and PB2 should be improved to reveal the evolutionary dynamics of H5N1 viruses in mammals

Response: Thank you. The purpose of the phylogenetic trees was not to reveal dynamics of evolution of H5N1 in mammals. We analysed viral protein sequences from cetaceans hosts with co-occurring isolates to investigate spill over to cetaceans. The evolutionary dynamics of H5N1 viruses in mammals has been extensively studied, for instance by Pardo-Roa et al, 2025 (https://doi.org/10.1038/s41467-025-57338-z).

Comments of Reviewer 2: Although the map with the locations of H5N1 viruses in cetaceans is provided in figure 2, the global Migratory Flyway in figure 2 is recommended to support the relationships of migratory seabirds and cetaceans infections. The distribution of avian influenza viruses has close relationship with the migratory Anseriformes and Charadriiformes (https://doi.org/10.1016/j.jia.2023.12.033).

Response: Thank you for your suggestion. We have moved figure 2 to the results section and re-numbered it as figure 1. We believe that in this introduction context, including flyways in the figure would distract the reader from the aimed message to provide in the results section. Also, the migratory pathways of relevance to this work would have to include a lot of overlapping areas related to the movement of cetaceans and seabirds that would distract the reader from the main message.

Comments of Reviewer 2:

In the discussion section, there are large numbers of words to discuss the H5N1 viruses, however, there no word to discuss prevention of H5N1 by vaccine. Vaccination of highly pathogenic H5 viruses in domestic have been proven the most effective way to prevent global circulation of H5 (PMID: 34757542). Discussion of vaccination of H5N1 viruses is recommended in discussion.

Response: Thank you for your comment. We have introduced the sentence “Reducing H5N1 outbreaks in wild birds through vaccination of poultry has been pro-posed as a strategy to prevent global viral dissemination [6, 81]” (page 11, lines 385-386).

Reviewer 3 Report

Comments and Suggestions for Authors

This manuscript presents a timely and comprehensive review of influenza A virus (IAV) infections in cetaceans, with particular emphasis on the emerging threat posed by clade 2.3.4.4b H5N1 highly pathogenic avian influenza viruses. The authors successfully summarize the historical context, geographical distribution, and molecular characteristics of IAV strains detected in cetaceans, highlighting important mammalian-adaptive mutations in viral proteins such as HA, PB2, and NP. The manuscript is generally well structured and offers valuable insights into the ecological and evolutionary dynamics of IAV at the wildlife–ocean interface.

Author Response

Response to Reviewer 3 Comments

Thank you for your comments and time dedicated to review our manuscript.

We have introduced major changes including reorganization of several sections of the manuscript. Also, we have improved figure quality and English.

In the introduction, the basic knowledge of influenza virus has been summarised, and paragraphs 1 and 2 of the introduction have been merged in a single paragraph (page 1, lines 34-42).

Part of the sentences previously included in section 3.1 of Results explaining in detail references compiled in Table 1 have been moved to the last paragraph of the Introduction.

In Results Section 3.1, changes have been introduced so that it is now exclusively dedicated to refer to the conclusions after compiling all the reviewed literature concerning the IAV subtype, the cetacean species infected by IAV, and the marine ecosystem (location). Also previously named figure 2 is now figure 1, and it is included in the Results section 3.1.

In the discussion, several sentences have been deleted or moved to the introduction or the results section.

Please find revisions in track changes in the re-submitted files.

Round 2

Reviewer 1 Report

Comments and Suggestions for Authors

N/A